# Training Robots to Evaluate Robots: Example-Based Interactive Reward Functions for Policy Learning

**Kun Huang**     **Edward S. Hu**     **Dinesh Jayaraman**
GRASP Lab, University of Pennsylvania
{huangkun, hued, dineshj}@seas.upenn.edu

**Abstract:** Physical interactions can often help reveal information that is not readily apparent. For example, we may tug at a table leg to evaluate whether it is built well, or turn a water bottle upside down to check that it is watertight. We propose to train robots to acquire such interactive behaviors automatically, for the purpose of evaluating the result of an attempted robotic skill execution. These evaluations in turn serve as "interactive reward functions" (IRFs) for training reinforcement learning policies to perform the target skill, such as screwing the table leg tightly. In addition, even after task policies are fully trained, IRFs can serve as verification mechanisms that improve online task execution. For any given task, our IRFs can be conveniently trained using only examples of successful outcomes, and no further specification is needed to train the task policy thereafter. In our evaluations on door locking and weighted block stacking in simulation, and screw tightening on a real robot, IRFs enable large performance improvements, even outperforming baselines with access to demonstrations or carefully engineered rewards. Project website: https://sites.google.com/view/lirf-corl-2022/

## 1   Introduction

Consider a kitchen robot that must perform a large number of tasks, such as opening a refrigerator, cutting vegetables, tightening a water bottle lid, or flipping a pancake. How might this robot acquire these skills? Common skill acquisition approaches involve heavy engineering and expertise for each skill, directed either towards developing model-based control policies, or towards specifying dense reward functions for reinforcement learning (RL). These approaches do not scale well to the goal of acquiring large numbers of skills. Consequently, there is growing interest in scalable approaches for RL-based skill acquisition that permit easy task specification by non-experts.

A particularly promising task specification framework that we call exemplar rewards, involves specifying tasks merely by showing the robot learner what the environment should look like after a well-executed skill [1, 2, 3, 4, 5, 6, 7]. For example, to specify the task of opening a refrigerator door, it would suffice to show the learner some images of open refrigerators. Then, once the skill is specified, the robot performs RL with the aim of altering the environment to generate observations similar to those examples. When this framework is successful, a human user teaching the robot does not need technical expertise and does not need to even demonstrate full behaviors for task execution. Instead, they need only capture images of the outcome.

Although this framework is appealing, many tasks cannot be specified by image observations of task success, or any other fixed sensor setup. For example, consider the task of closing a water bottle, as in Figure 1. What image might specify a successful execution of this skill? An image of a water bottle with a slightly loose cap would look identical to one with a tight cap, but a bottle closing policy that does not fully tighten the cap would be useless. Thus, in this setting, an observation of the task success state from a camera constitutes only a partial observation and does not suffice to specify the task. Specifically, when non-success states might look visually similar to success states, image-based task specification and assessment fails. Figure 1 shows a few examples of this state aliasing problem: Is the bottle cap tight? Is the door locked properly? Is the tower of objects with unknown masses stable? There are many more: Is a screw tight? Is the leg of the table secured well? Does the smoothie have the right consistency? None of these questions can be answered correctly from passive image observations.

6th Conference on Robot Learning (CoRL 2022), Auckland, New Zealand.

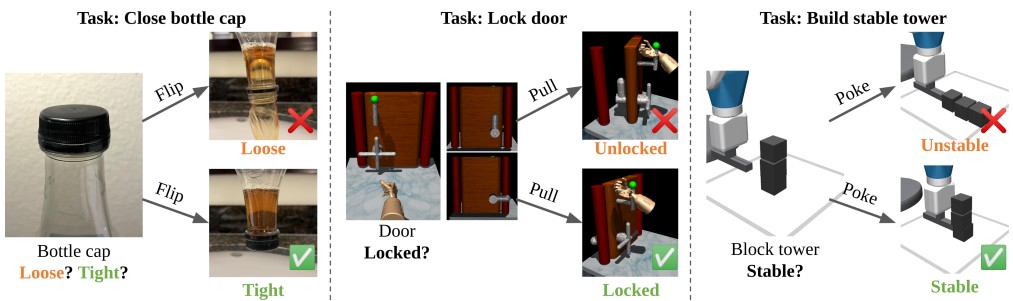

Figure 1: In each of these three settings, task success / failure cannot be determined from passive image observations, but is easily revealed through appropriate interactions.

The failures of passive image-based assessment in these settings preclude robotic skill learning from image examples of task success. To fix this while retaining the versatility and user-friendliness of the exemplar rewards framework, we propose an *interactive* approach for evaluating task success. As motivation, consider how a human might turn a water bottle upside down to confirm that it does not leak, and is therefore closed well. Here, the interaction reveals the key determinant of task success, namely, whether or not the cap is watertight.

We design an approach to automatically learn such interaction behaviors to evaluate task success. Our examples are no longer mere images, but instead *real actionable physical instances*. In the water bottle setting, rather than image snapshots of closed water bottles, our approach receives actual physical bottles with tightly closed caps. The robot can interact with these objects, such as by turning them upside down, to discover what about them constitutes successful execution of the desired "bottle closing" skill (i.e., the bottle does not leak) and how they behave differently from the outcomes of unsuccessful execution (i.e., it leaks). In the other examples in Figure 1, a robot might try to pull the door open or nudge the tower to evaluate whether the task is complete.

We make two key contributions. First, we propose an approach to train *interactive reward functions*, which are policies that specify what it means to correctly perform a skill and thus drive the improvement of the task policy learners in a reinforcement learning framework. Next, we show how those same reward policies can be reused to enable *in-the-loop introspective verification* behaviors during online skill execution, making robots more scrupulous and reliable. We evaluate these contributions on three simulated and real robotic control settings, and show substantial improvements in task policy learning and execution.

## 2 Background and Problem Setup

### 2.1 Classifier-Based Exemplar Rewards in fully observed MDPs

Reinforcement learning (RL) methods offer the promise of scalable data-driven synthesis of robot controllers for arbitrary new tasks. Consider robotic task settings formalized as Markov decision processes (MDPs), defined by the tuple $(\mathcal{S}, \mathcal{A}, \mathcal{T}, R, \mu)$. At each time $t$, an agent selects an action $a_t \in \mathcal{A}$ in state $s_t \in \mathcal{S}$, transitions to the next state $s_{t+1}$ with probability $\mathcal{T}(s_{t+1}|s_t, a_t)$, and receives reward $r_t = R(s_t, a_t, s_{t+1})$. Thus, the agent emits actions into the environment and receives two things in return: new state observations and rewards. Ignoring discounting, a good RL task policy $\pi_T(a_t|s_t)$ selects actions to maximize the sum of rewards over time $\sum_t r_t$, starting at a state drawn from an initial state distribution $\mu$.

Thus, it is the reward function $R$ that specifies what task to perform in a given environment, and RL approaches in practice require expertise and effort for reward engineering [8] to specify each new task, avoiding misspecification and guiding efficient learning. This often takes the form of tuned weighted combinations of many carefully constructed heuristic terms in the RL reward objective [9, 10, 11, 12], or even privileged task-specific sensors added to the environment during training [13, 14]. To circumvent laborious manual reward design, many methods aim to learn rewards from data. Inverse reinforcement learning methods [15, 16, 17, 18] learn task reward functions from optimal demonstrations, but such demonstrations are typically expensive and may even be impossible to obtain. Other methods train RL agents by learning rewards explicitly or implicitly from interactive human feedback [19, 20, 21, 22, 23], but these have the drawback of requiring in-the-loop queryable human teachers.

We build upon the popular "exemplar rewards" framework, which explores task specifications purely through examples [1, 2, 3, 4, 5, 6, 7]. In classifier-based exemplar rewards approaches [1, 3], the human teacher trains a classifier $\hat{R}(s)$ to label task outcomes as successes or failures, by gathering some success examples manually, and then generating failure examples from the task policy as it trains. Armed with this learned reward function $\hat{R}(s)$, the agent trains its task policy $\pi_T(a|s)$ to generate success outcomes and thereby maximize the estimated rewards. An appealing property of this framework is that training a policy for each new task only requires a specification of the final goal state that should be achieved, without any need to specify how to achieve it.

Our work extends this exemplar rewards framework: as motivated in Section 1, rather than relying on passive observations, we will train interactive evaluation policies to evaluate task success, and these evaluations will in turn serve as "interactive reward functions" for training task policies. This permits applying exemplar rewards to "partially observed" settings that do not permit accurate task success evaluation from images alone. We expand on the partial observability problem below.

## 2.2  Partial Observability, State Aliasing, and History-Based POMDP Policies

Why are classifier-based exemplar rewards approaches not suited to settings such as the door locking task in Figure 1 (middle)? The root cause is "partial observability." To solve this task, the robot must push the door into the closed position and then rotate the latch into the locking position. It only sees images from a fixed camera in front of the door. The latch, however, is occluded, and the robot can only observe and act upon the four handles on the front of the door, which rotate the latch through an axle mechanism. The handles are visually identical, so any visual configuration of the handles corresponds to any one of four configurations of the latch behind the door. While this setting is constructed as a pedagogical example to expose and study the state aliasing problem, it is representative of several real-world settings, as discussed in Section 1 and Figure 1.

Such settings are modeled as partially observable MDPs, or POMDPs [24]. At each time, the agent only partially observes the underlying latent task state $s_t$ (i.e. the latch configuration), through an observation $o_t \sim O(o|s_t)$, where the observation function $O$ generates samples from the observation space $\Omega$ (i.e, images of the handle configuration). The original task state $s_t$ is not directly observable. Thus, the task POMDP is defined by $(\mathcal{S}, \mathcal{A}, \mathcal{T}, R, \Omega, O, \mu)$.

Several prior works deal with the question of how the task policy in a POMDP might map to good actions despite only seeing partial observations at each time. The key to this is the assumption, very often reasonable, that while individual observations are indeed incomplete, observation *histories* contain much more information about the state. Then, a POMDP policy can perform well by relying on a history of observations and actions, either directly [25, 26, 27, 28] as $\pi_T(a_t|o_{t-H:t}, a_{t-H-1:t-1})$, or by first estimating a belief distribution $b_t(s_t|o_{t-H:t}, a_{t-H-1:t-1})$ over the state and then mapping to actions [24, 29, 30, 31, 32], as $\pi_T(a_t|b_t)$. In the door locking setting, the robot might observe the latch position before closing the door. Therefore, even if the latch is occluded after the door is closed, a history-based policy could still track its position. In other words, if trained well with the right reward function, such history-based POMDP policies can still learn good task behaviors.

## 3  Exemplar Interactive Reward Functions

This brings us to the key question: how to specify reward functions $R$ for training POMDP policies? In particular, could we extend the appealing exemplar rewards framework to POMDPs?

First, note that partial observability has to do with what *policies* can see, not where *rewards* come from. Therefore, it is feasible in theory to provide state-based exemplar rewards $R(s)$ that are not limited by partial observability. This implies relying on additional sensors for reward estimation during training that are not available to the policy $\pi_T$ that is being trained. In other words, while the states $s_t$ are no longer observable to $\pi_T$, it may still be possible to learn exemplar rewards $\hat{R}(s_t)$ through privileged instrumentation of the environment during training to allow the reward function to observe the full state $s_t$. For example, privileged thermal cameras could measure fluid levels to train an RGB image-based fluid pouring policy [13]. However, such additional instrumentation is labor-intensive and task-specific, and sacrifices the appealing scalability and ease of use of the exemplar rewards framework.

How might we do this? Learning exemplar rewards $\hat{R}(o_t)$ based on the agent's instantaneous observations would suffer from state aliasing: in the door locking setting, such a reward function would not be able to distinguish a correctly latched door from one in which the latch is off by a factor

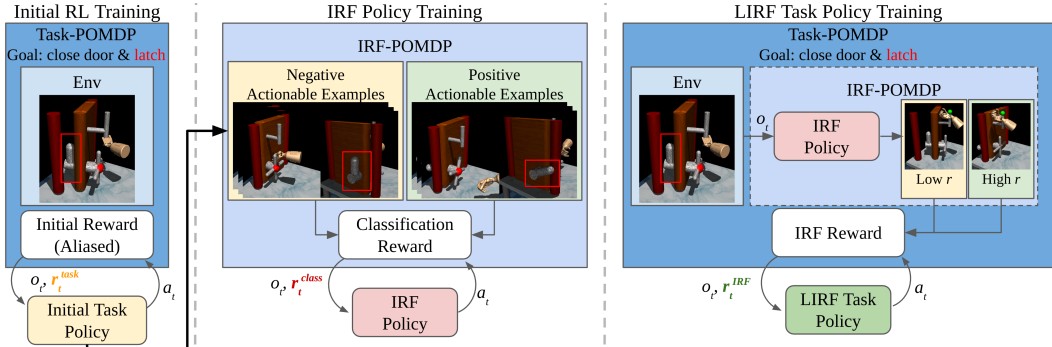

Figure 2: **Method Overview.** In a partially observable environment, (left) we first learn an initial task policy $\pi_T^0$ using passive classifier-based rewards. (middle) Then we train an IRF policy $\pi_R$ to distinguish between provided "actionable positive examples" and $\pi_T^0$-generated negative examples. (right) Finally, we use $\pi_R$ to provide the correct rewards for training a LIRF task policy $\pi_T$.

of 90°! Nor does the exemplar reward learning framework permit relying on observation histories, like the solution we described above for POMDP *policies*. To see this, recall that exemplar rewards are trained from samples of successful task *outcomes* rather than full demonstrations, so there is no observation-action history information available for training $\hat{R}(o_{t-H:t}, a_{t-H-1:t-1})$.

### 3.1    Actionable Examples and Interactive Reward Functions (IRFs)

Our primary contribution is a solution to this conundrum: rather than presenting successful outcome examples as singular pre-recorded observations, we propose to present them as *"actionable examples"*, with which the robot can interact and generate new observations. In other words, to learn a reward function for locking a door, the robot gets access to several physical instances of successfully locked doors. Then, instead of training passive reward functions from image observations, we propose learning *"interactive reward functions"* (IRF), which consist of robot action policies $\pi_R$ that reveal the task rewards. See the schematic in Figure 2. We expand upon this idea below.

First, we gather "actionable success examples" $P = \{\sigma_1^+, ..., \sigma_K^+\}$ and "actionable failure examples" $N = \{\sigma_1^-, ..., \sigma_K^-\}$. $P$ and $N$ contain object or environment configurations: in Figure 1, each $\sigma^+$ might be a tight bottle cap, a well-locked door, a stable tower, and so on. Now, the IRF policy may be seen as the solution to a new "IRF POMDP". This is identical to the task POMDP, except for two key differences. First, it is initialized to an actionable example state $s_0 = \sigma \sim P \cup N$. Next, its reward function $R$ is based on discovering the label of $s_0$: was it drawn from $P$ or from $N$? All other elements $(\mathcal{S}, \mathcal{A}, \mathcal{T}, \Omega, O)$ of the task POMDP are retained.[1]

Algorithm 1 shows pseudocode for our "Learning from IRFs" (LIRF) approach. We point out three key features here. **(1)** First, we bootstrap the task policy $\pi_T$ by training against a single-observation-based reward classifier $D(o_t)$. This corresponds to running prior exemplar reward approaches; we use VICE [1]. **(2)** Next, in keeping with the exemplar rewards framework, we provide only the positive examples $P$. Negative examples $N$ are instead generated by the above-initialized task policy. Thus, the IRF policy $\pi_R$ serves as a GAN-like adversarial critic [33, 18] for training the task policy $\pi_T$. **(3)** Finally, how could we discover the true task completion reward, i.e., the label $P$ (corresponding to positive reward) or the label $N$ (negative reward) at the end of IRF execution? For this, we could, in theory, train a classifier from the full IRF trajectory history. Instead, we find that it suffices to reuse the single-observation classifier $D(o_t)$ from (1) above, and apply it only to the final state after executing $\pi_R$. Intuitively, $\pi_R$ learns to modify the environment state $\sigma \xrightarrow{\pi_R} s^*$ such that $o^* \sim O(s^*)$ reveals whether $\sigma$ was drawn from $P$ or $N$. For example, in door locking, $\pi_R$ might learn to tug at the door: if the door is correctly locked ($P$), it stays closed, and if it is not locked ($N$), it opens. Thus, a single post-IRF observation suffices to classify $\sigma$. Figure 2 shows a schematic.

---

[1]In some experiments, we use different actions $\mathcal{A}$ for the IRF policy $\pi_R$ than for the task policy $\pi_T$.

---

**Algorithm 1** Learning from Interactive Reward Functions (LIRF) Framework

**Require:** a set $P$ of positive actionable examples that specify successful task execution.
1: Following image-based exemplar rewards approaches such as VICE [1], train a single-observation reward $D(o) \to [0, 1]$ and an initial task policy $\pi_T^0$ in the task POMDP.
2: Rollout $\pi_T^0$ for $n$ times, collecting negative actionable outcomes in $N$.
3: Train an IRF policy $\pi_R$ in the IRF POMDP, where the environment is initialized from state $\sigma \sim P \cup N$, to maximize the classification reward:

$$\mathbb{E}_{s_0 \in P} \left[ \sum_t \log(D(o_t)) \right] + \mathbb{E}_{s_0 \in N} \left[ \sum_t \log(1 - D(o_t)) \right] \tag{1}$$

4: Train the final LIRF task policy $\pi_T$ in the task POMDP, modified to include the additional terminal state reward $\hat{R}(o_T) = D(o_T) + \lambda D(o^*)$, where $o_T$ is the observation for the terminal state $s_T$, and $o^*$ is generated by the IRF policy $\pi_R$ as $s_T \xrightarrow{\pi_R} s^*, o^* \sim O(s^*)$. The final reward function uses the dense single-observation reward for intermediate states, and the terminal state reward for the final state.

$$\hat{R}(o_t) = \begin{cases} D(o_T) + \lambda D(o^*) & \text{if } t = T \\ D(o_t) & \text{otherwise} \end{cases} \tag{2}$$

---

### 3.2 IRFs as Verification Mechanisms for Introspective, Fault-Tolerant Behaviors

Next, we observe that IRFs $\pi_R$ rely on no privileged information beyond the task policy $\pi_T$'s own observations. Thus, while their primary purpose is to provide reward functions for training task policies, IRFs can also be deployed together with the task policies after training. We propose a simple approach to use IRFs as verification mechanisms *in-the-loop* during task execution. Once the task policy $\pi_T$ is executed for a fixed episode length, we will run the IRF $\pi_R$ to determine whether the task is complete. If it is detected as incomplete, we will resume executing the task policy. This perform-verify cycle continues until the IRF is satisfied that the task is complete (or timeout). Appendix A.1 contains pseudocode for this procedure.

## 4 Other Related Work

LIRF trains task policies through rewards computed by an interactive reward function, which is itself a robot policy trained from examples. While Section 2 extensively discussed connections to reward design, exemplar rewards, and POMDP policy learning to set up our approach, we now position our contributions against two other key related areas of prior work.

**Adversarial Robot RL.** Like LIRF, some prior works train a target task policy against another "adversary" policy [34, 35, 36, 37]. Pinto et al. [37] train a grasping robot to select more stable grasps by competing against an adversary robot that tugs at the grasped objects to dislodge them. Rather than adversary policies that destabilize the task policy to increase robustness, we train adversary policies that provide outcome-based rewards to train the task policy.

**Interactive Perception and Verification.** Our method draws on ideas from interactive perception [38], where agents in partially observed settings act to obtain information about the latent state. Interactive task verification mechanisms (see [39] for a survey) employ human-specified interactive perception behaviors to assess the state of a task. For example, a robot may use engineered motion primitives or perturbations to detect and correct task failures in manipulation [40, 41]. In contrast, we *learn* interactive verification behaviors from examples, and use them not only during task execution, but also to define the reward function for training task skill policies. One way to frame our contribution here is as follows: in reinforcement learning, agents emit actions into the environment and receive in return not just new states, but also *task rewards*. To our knowledge, LIRF is the first approach that employs interactive perception to directly estimate task rewards, rather than the state.

## 5 Experiments

We design various simulated and real-world experiments to evaluate LIRF for example-based task specification in partially observable tasks that manifest the state aliasing issue. We aim to answer the following questions: (1) Does our interactive reward learning framework enable us to learn policies to solve partially observable tasks that alternative approaches cannot, with comparable or even greater task supervision? (2) Does the learned IRF policy sensibly evaluate task completion and provide

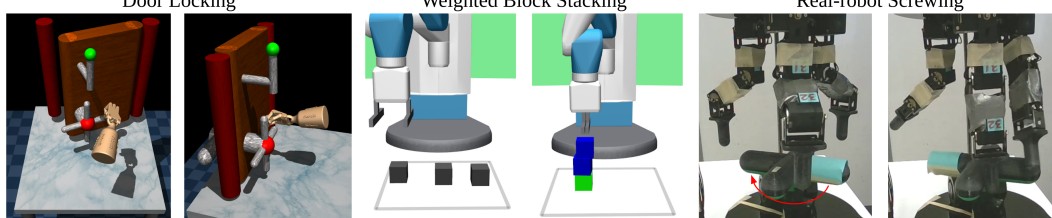

Door Locking       Weighted Block Stacking       Real-robot Screwing

Figure 3: Our three tasks. For each pair of 2 images, we show the task initialization on the left, and the goal state on the right. In the middle panel, the 3 blocks are visually identical but they differ in weight. The heaviest block is green for visualization. In the right panel, we put a blue mark on the valve for visualizing the orientation.

good rewards? (3) Can using the IRF policy in-the-loop for task verification improve task execution performance?

**Task Setups.** We evaluate LIRF on 3 tasks, illustrated in Figure 3, and described below. See Appendix A.2 for more details and the project website for experiment videos.

- **Door Locking (Sim):** We instantiate the door locking task from sections 2.2 and 3 with an Adroit Hand using MuJoCo [42]. The robot has a fixed viewpoint in front of the door, and must close the door and rotate the symmetric, four-handled doorknob to fully lock it (see Figure 3). The observation space mimics perfect vision: it consists of the door orientation, the door frame position, the *aliased* cross-handle orientation $[0, 90°)^2$ and the Adroit hand pose. The door is initialized as open, and the position of the door frame and the orientation of the latch are randomized.

- **Weighted Block Stacking (Sim):** We further test our algorithm on a weighted block stacking task based on the MuJoCo Fetch environment [43] as shown in Figure 3. The goal of this task is to stack 3 visually identical blocks into a "stable" tower. One block significantly heavier than the other two blocks, so the optimal strategy is to place it at the bottom. Again, the observation space mimics perfect vision: only the unordered poses of blocks are observable. The weight of a block is only revealed after it is picked up, to mimic physically plausible "hefting" behavior.

- **Screw Tightening (Real):** To examine the robustness and generality of our algorithm, we test it on a real-robot screwing experiment using a D'Claw [44] that has 9 joints (Figure 3). The objective of this task is to turn the 4-prong valve clock-wise for around 180° into the "tightened" state (white line on the valve base). We engage a motor underneath the valve to mimic screw locking. Again, the observation consists of the historical aliased valve angles, i.e. $[0, 90°)$ and robot joint angles.

**LIRF Implementation Details.** We use the soft actor-critic (SAC) algorithm [45] to train both the task policies and the IRF policies for each experiment and report results with mean and variance over 5 seeds for each simulated task. The weight $\lambda$ for the sparse IRF reward in Eq 1 needs to be "large enough" (Appendix A.6) to have a substantial effect on LIRF training, but beyond this, the algorithm is not very sensitive to $\lambda$; we set $\lambda = 1000$. All policies are trained until convergence, which takes around 2M simulation steps for door locking and block stacking LIRF policies, 50k steps for screwing LIRF policies ($\sim 10$ hours on the real robot), and around 20k steps for all IRF policies. More details in Appendix A.5. We will release all code and environments.

**Baselines.** For evaluating our LIRF algorithm, we compare against RL policies learned with VICE exemplar rewards [1] and with ground truth state-based rewards ("GT State Reward"). In addition, we compare against GAIfO [46], an imitation-from-observation algorithm that learns from demonstrations. We provide as many demonstrations to GAIfO as the number of episodes with positive examples provided to LIRF (namely, 250, 10k, and 100 for door locking, weighted block stacking, and screwing respectively). Note that LIRF actually uses fewer positive examples than the number of positive episodes, (110, 7K, and 1 actionable positive examples respectively) since the same example can be reused across episodes if it is not destroyed. Finally, when feasible, we engineer an interactive reward function policy (Manual IRF) that performs hand-coded actions to evaluate the task: hand-coded unscrewing behavior for screwing, and a random poking action for the block

---

[2]The cross-handle is symmetrical and looks visually identical at 90°offsets, e.g. 10°, 100 °, 190 °, 280 °

| | Task Specification | Door Locking | Block Stacking | Screwing |
|---|---|---|---|---|
| VICE [1] | Goal Examples | $0.032 \pm 0.015$ | $0.310 \pm 0.056$ | 0.02 |
| LIRF (Ours) | Actionable Goal Examples | $\mathbf{0.640 \pm 0.053}$ | $\mathbf{0.846 \pm 0.027}$ | 0.82 |
| LIRF+Verify (Ours) | Actionable Goal Examples | $\mathbf{0.958 \pm 0.007}$ | $\mathbf{0.884 \pm 0.018}$ | **0.99** |
| GAIfO [46] | Demonstrations | $0.112 \pm 0.024$ | $0.275 \pm 0.043$ | 0.0 |
| Manual IRF | Human Engineering | - | $0.613 \pm 0.058$ | 0.85 |
| GT State Reward | State+Human Engineering | $0.714 \pm 0.047$ | $0.956 \pm 0.021$ | 0.87 |

Table 1: Task success rates of our method and baselines on the three tasks.

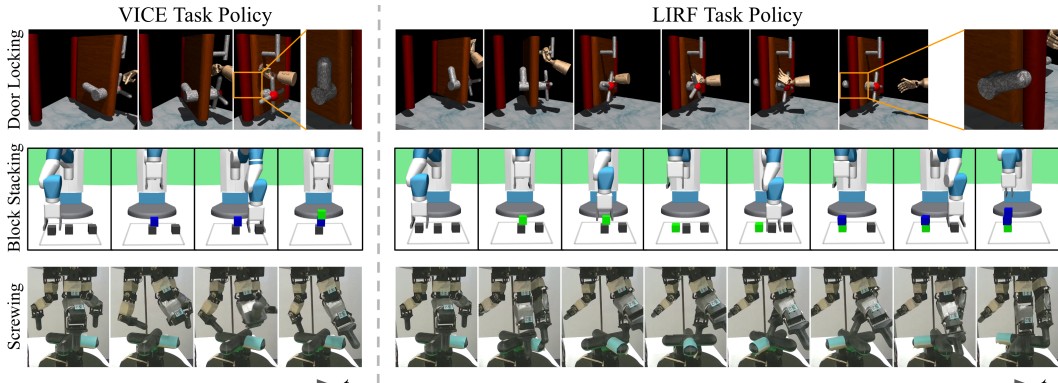

Figure 4: LIRF task policy rollouts (right), compared to VICE (left). Note row 2, where VICE builds an unstable tower with the heavy block on top (visualized in green here, but this is not observable to the agent). LIRF self-corrects mid-episode to build the tower with heavy block on bottom. See Supp slides for more video examples.

stacking.[3] To isolate the effect of reward functions, all methods share the same input observation space for their policies. See Appendix A.4 for more details on the baselines. Note that GAIfO, GT State Reward, and Manual IRF all involve more painstaking and comprehensive task specifications than LIRF, but offer useful comparison points nevertheless.

## 5.1 Results

**Does LIRF succeed in partially observable settings?** Table 1 shows task success rates for our approach and baselines. LIRF outperforms VICE across all tasks, showing that LIRF successfully extends the exemplar reward framework into partially observable settings where prior exemplar rewards methods like VICE fail. LIRF also performs on par or better than baselines that use additional / more expensive forms of task supervision, such as demonstrations, human engineering, and privileged sensing of ground truth state.

Qualitatively, LIRF policies show good task-solving behaviors for each task - they learn to lock the door, construct towers with the heaviest block at the bottom, and fully tighten the screw (see Figure 4). LIRF even demonstrates sophisticated self-correction behavior in the block stacking task by rebuilding the tower if the heavier block is incorrectly stacked on top of a lighter block.

Among baselines, VICE and GAIfO perform poorly due to state aliasing, and LIRF is competitive or superior to Manual IRF, showing that *learning* an IRF policy with RL may yield better rewards for training partially observable task policies.

**Do the IRF policies $\pi_R$ sensibly evaluate task completion?** First, we evaluate the success rate of the IRF policy $\pi_R$ at distinguishing between positive and negative examples for each task in Table 2. IRF-based task success classification is highly accurate for all tasks. Figure 6 shows examples of learned reward function policies. The IRF policies learn sensible strategies to separate visually indistinguishable positives and negatives. In the door locking task, IRF pulls the fixed handle; in stacking, it pokes

| Task | Accuracy |
|---|---|
| Door Locking | $0.992 \pm 0.006$ |
| Block Stacking | $0.852 \pm 0.023$ |
| Screwing | 0.98 |

Table 2: IRF task success classification accuracies.

---

[3]For Adroit hand door locking, the action space is too large to reasonably hand-code any IRF behavior.

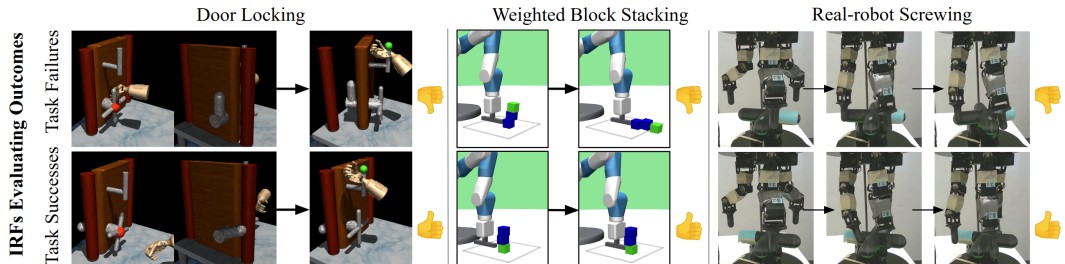

**IRFs Evaluating Outcomes**

Task Failures · Task Successes

Door Locking · Weighted Block Stacking · Real-robot Screwing

Figure 6: Running the IRF policy $\pi_R$ on the success and failure actionable examples. Left (Door Locking): $\pi_R$ grasps the door and pulls to verify locking. Middle (Block Stacking): $\pi_R$ checks for tower stability by gently poking the bottom block. Right (Screwing): $\pi_R$ attempts to lightly unscrew - the screw in the top row rotates as it is not tight, whereas the bottom-row does not.

the bottom block; in screwing, it applies a small counterclockwise torque to unscrew the valve. Finally, IRF policy learning is quite efficient in terms of the number of actionable positive examples required. Door locking, screwing, and weighted block stacking need 110, 1, and 7k actionable positive examples respectively (note that the same positive example can be reused for multiple IRF training episodes, unless its "positiveness" is destroyed at the end of that episode). Block stacking is the hardest, since distinguishing stable from unstable towers requires a finely tuned poking force and motion, and even "stable" block tower examples are prone to toppling quite easily. See Appendix A.7 for more details.

**Can the IRF policy be used for in-the-loop verification?** As seen in Table 1, LIRF+Verify, which reuses the learned IRF policy during task execution, outperforms all baselines, even beating the ground truth reward baseline in door locking and screwing. Figure 5 shows how LIRF+Verify improves performance with more perform-verify iterations. LIRF+Verify makes the biggest difference in door locking: here, plain LIRF fails mainly because the policies operating on partial observations do not learn precisely how much to turn the latch without the aid of $\pi_R$ to declare task completion. This is also why GT State Reward performs quite poorly on this task despite training

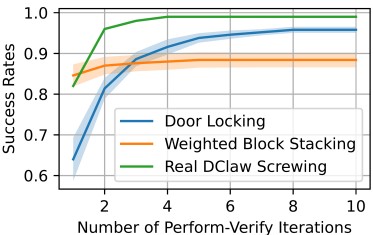

Figure 5: LIRF+Verify success rates vs. perform-verify iterations.

on the true state-based reward function (see also Appendix A.8). LIRF+Verify's gains over LIRF are smaller in block stacking and screwing, because LIRF failures here occur from low-level imprecision in pick-and-place or valve-turning. Here, LIRF+Verify helps by allowing the task policy $\pi_T$ to try again when it fails. We show video examples on the project website.

## 6 Conclusions

We have presented LIRF, a framework for conveniently training example-based interactive robot policies to evaluate robot task policies, in order both to provide rewards to train them, and to verify their execution in partially observed tasks. While our results show substantial improvements over baselines, LIRF currently has two drawbacks. LIRF is best applied to settings where the initial task policies trained from single-image-based rewards fail most of the time. When this is not the case, such as in fully observable settings, IRF policies yield marginal or no benefits over the initial policies (Appendix A.9), and therefore may not be the most frugal algorithmic choice. Next, LIRF requires the physical storage of "actionable outcomes" as positive examples. While the number of such examples in some tasks may be small enough to not be a major concern (e.g., among our experiments, 1 for screwing, 110 for door locking), in other cases, storing and presenting a large number of positive example objects to train the IRF might be cumbersome (e.g., 7k for block stacking). We discuss scalability in more detail in Appendix A.10. Future work overcoming these limitations could further expand the domain of applicability of LIRF.

## 7 Acknowledgement

This work was supported by the U.S. Office of Naval Research under grant ONR N00014-22-1-2677, and a gift from NEC Laboratories America.

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
