# OpenReview forum: "Training Robots to Evaluate Robots: Example-Based Interactive Reward Functions for Policy Learning"
_robot-learning.org/CoRL/2022/Conference — CoRL 2022 Oral_

### Official Review · Reviewer_T7V5 · 2022-07-30

**Originality:** Very Good
**Technical Quality:** Very Good
**Clarity Of Presentation:** Excellent
**Impact:** 3

**Recommendation:**

Strong Accept: I recommend accepting the paper and will argue for my recommendation even if other reviewers hold a different opinion.

**Summary:**

The paper proposes a framework for policy learning in a partially observable setting. In this setting, the observation itself does not contain full information about the Markovian state, and hence, does not have sufficient information to determine the ground truth reward or task success. To approach this, the paper proposes to train another policy, called the IRF policy, to perturb the final state in a way such that task success is transferred to the observation space. Therefore, this information can then be used as a reward function to train the task policy, or to verify whether the task is indeed achieved.

The proposed framework is tested on 2 simulated tasks and 1 real robot task. In all cases, the proposed framework is shown to significantly outperform baselines that 1) only uses observation for exemplar rewards and 2) directly imitate the demonstrations. However, it seems like the proposed framework is only evaluated on tasks that VICE achieves a low success rate.

**Issues:**

See **Strengths and Weaknesses**.

**Quality Of The Limitations Section:**

Limitations are addressed clearly

**Reviewer Expertise:**

4: The reviewer is confident but not absolutely certain that the evaluation is correct

**Robotics Focus:**

Sufficient demonstration on hardware

**Strengths And Weaknesses:**

### Strengths
- **Organization**: The paper is very well written and organized. The motivation of the paper is clearly stated in the introduction. The authors use concrete examples, both from real life and the actual experiments to illustrate the intuition behind the proposed idea. The proposed algorithm is well-stated and easy to follow. The experiment section provides clear description on each of the tasks, and detailed information about their setup.
- **Literature survey**: The authors clearly stated the limitation of existing work in exemplar rewards: They can not be directly applied to the partially observable setting since the task may falsely appear to have succeeded. The authors also discuss how their work differs from existing literature in Adversarial Robot RL and Interactive Perception and Veriﬁcation.
- **Technical approach**: The technical approach is well motivated and is built upon a sound intuition. The explanation of the algorithm is thorough and clear.
- **Experiment validation**:  The proposed framework is tested on 2 simulated tasks and 1 real robot task. In all cases, the proposed framework is shown to significantly outperform baselines that 1) only uses observation for exemplar rewards and 2) directly imitate the demonstrations. The authors provided thorough information on the tasks, and how the policies are trained and evaluated. The experiments verified that the proposed framework is indeed critical in the partially observable setting.
- **Final discussions**: It is appreciated that the authors discussed some of the limitations of their work in a fair way in the conclusion session.

### Weaknesses
- **Assumptions on VICE performance**: As noted by the authors, the proposed approach assumes that the task policy initially performs poorly on the task. (If I understood it correctly, all the examples generated by VICE are considered negative, is that right?). It seems like a limitation since VICE policy may already be quite good in some tasks, e.g. if we only need to close the door. Can the authors provide more information on what will happen in this case? Does the performance of the task policy degrade after LIRF, or does it just stay roughly the same? Maybe the authors can, in the appendix, provide an experiment where VICE is sufficient to solve the task, and show what happens after applying LIRF. To follow up, does it imply that LIRF will fail to improve (or even degrade) performance if the initial task policy success rate is high enough? If so, how do we determine whether we should use LIRF or just use the initial VICE policy? It seems hard since the paper assumes no access to the ground truth reward function.

- **Only training IRF once**: The authors noted that the previous limitation is now preventing iterative training of IRL policies, which makes sense. However, I am curious whether this current framework can encourage the task policy to fail at tasks in a “creative” way that can not be caught by the IRF policy. Since the IRF policy is only trained once with the VICE policy, it will not be able to adapt to other failure modes that the updated task policy induces. Although it seems like it is not the case in the experiments shown in the paper, I am wondering if the authors have observed such a phenomenon at all, and if the authors have any intuition on what to do in such cases. Minor point: Since the IRF policy is only trained once, it would be a little misleading to call it “GAN-like adversarial critic”.

- **Size of N**: What is the size of $N$ for the current experiments?
- **Choice of hyperparameter**: How sensitive is LIRF to the choice of $\lambda$?
- **Block stacking**: From the quantitative results, as well as the number of positive examples needed, it seems like the block stacking experiment is most challenging. However, according to the appendix, the authors already made various attempts to simplify it, such as using motion primitives, and provide weight information once a block is picked up. Can the authors provide more explanation on why this experiment is still so challenging?
- **Screwing task success**: Can the authors provide more information on how the task success rate is evaluated in the screwing experiment? It is not very clear what does “the valve reaching the tightened state” exactly mean.
- **Typo?**: “The valve can rotate freely without limits when the motor is not engaged, whereas it is locked in place or controllable through the motor when the motor is not engaged.” Should it be “the motor is engaged” near the end?


**Summary Of Recommendation:**

The paper addresses a well-motivated problem of policy training in a partially observable setting. The paper is very well written and organized. The motivation of the paper is clearly stated in the introduction. The proposed algorithm is well-stated and easy to follow. The literature survey is thorough, The authors clearly stated how their work differs from existing work in the literature. The experiment section provides clear description on each of the tasks, and detailed information about their setup.  The authors also addressed clearly the limitations of their approach.

The major limitation, however, is that the current setup only seems to work when the initial task policy has poor results (see the first point in **Weaknesses**). Based on the assumptions in this paper, it is also unclear how to evaluate the initial task policy, as the ground truth state and hence reward is assumed unknown. The paper would have significantly greater impact if this limitation is addressed.

**Updates after the rebuttal period.** The additional results from the authors show that 1) LIRF will not degrade performance if VICE is already sufficient for solving the task; 2) multiple iterations of LIRF also does not degrade performance. This clears up my major concern. With that, I am glad to raise my rating from weak accept to strong accept.

---

> ### Author Response · Authors · 2022-08-18
> **Response to Review T7V5 (1/2)**
>
> We thank reviewer T7V5 for the effort spent on reviewing our manuscript and providing thorough feedback!
>
> > How do we determine whether we should use LIRF or just use the initial VICE policy? It seems hard since the paper assumes no access to the ground truth reward function.
>
> Note that LIRF is designed for the partially observable setting where passive goal specification methods like VICE naturally fail. The fact that LIRF is most valuable in a specific setting and may not yield improvements in others is not, in our eyes, a dealbreaker — we argue that it is quite reasonable to ask a user to recognize where LIRF will perform better than standard exemplar reward methods: namely, when an observation snapshot of a successful task outcome provides only an aliased description of task success, such that the observation can be achieved even without being actually successful at the task. For example, take a water bottle cap that looks the same regardless of whether it is tight or not: a picture of this water bottle constitutes an aliased specification of the task of water bottle cap closing.
>
> > Assumptions on VICE performance: As noted by the authors, the proposed approach assumes that the task policy initially performs poorly on the task. (If I understood it correctly, all the examples generated by VICE are considered negative, is that right?). It seems like a limitation since VICE policy may already be quite good in some tasks, e.g. if we only need to close the door. Can the authors provide more information on what will happen in this case?
>
> > … does it imply that LIRF will fail to improve (or even degrade) performance if the initial task policy success rate is high enough?
>
> Yes, all examples generated by VICE [1] are treated as negatives in our approach, just as they are treated as negatives within the VICE approach! We hypothesized in Sec 6 that IRF may not generate effective verification behaviors if the current task policy (a VICE policy or a previous iterate of LIRF policy when there are multiple iterations) is already sufficient. However, this does not mean that the final LIRF task policies will be worse in such cases: rather it means that gains will be minimal or zero in these settings. We show empirical evidence in the response to the next question, where we address iterative training.
>
> [1] Fu, Justin, et al. "Variational inverse control with events: A general framework for data-driven reward definition." Advances in neural information processing systems 31 (2018).
>
> > …Only training IRF once: The authors noted that the previous limitation is now preventing iterative training of IRL policies….
>
> It is not true that LIRF is incompatible with iterative training of the task and IRF policies against each other. We have tried iterative training in early experiments and found it to be only marginally beneficial, hence why we choose to do 1-iteration training for simplicity. For example, in an early version of our weighted block stacking task (simplified action space and state representation), we tried iterative training and recorded the success rate of the task policy and classification accuracy of the IRF policy after each iteration.
> | Policy | Task policy success  |
> |-------| -------|
> | VICE | 78% |
> | LIRF 1st | 95% |
> | LIRF 2nd | 95.1% |
> Here, we can see that the VICE task policy already has a relatively high initial success rate of 78%. After applying 1 round of training (i.e. training IRF policy, then training a new task policy against it), the task policy significantly improves to 95%. After another round the task policy plateaus at 95.1%. This shows that LIRF can handle good initial policies and that multiple rounds of training do not hurt LIRF. We emphasize again: in such cases, LIRF has diminishing returns for the added algorithmic complexity over plain VICE or single-iteration LIRF and may not be the most frugal algorithmic choice, but it **will not hurt task performance**.
>
> Time permitting within the response period, we will run an additional experiment to further evaluate LIRF in a setting where VICE already achieves near-perfect performance, such as a fully-observed state-based task setting like the door closing experiment suggested by the reviewer.
>
> Perhaps our original somewhat conservative phrasing of limitations in Sec 6 led to the reviewer overestimating their extent. We have rephrased our limitations section to more clearly circumscribe our limitations and avoid confusion. We attached the updated pdf in the response for the meta review.

---

> > ### Comment · Reviewer_T7V5 · 2022-08-27
> > **Thank you for your detailed response**
> >
> > Thank you for providing such a detailed response!
> >
> > I am glad to see that LIRF can achieve good results on tasks that VICE is sufficient, and also see that multiple-iteration of LIRF do not degrade results. The additional clarifications are also helpful and provide more intuitions on why this is the case. I understand that the rebuttal period is quite limited, and I hope that the authors can add new results to the final version of the paper.
> >
> > I also appreciate the authors test LIRF on different values $\lambda$. The results show that the proposed approach is not sensitive to the choice of $\lambda$ as long as it is large enough. The author also provide conniving explanation on why $\lambda$ needs to be large.  Again, it would be nice if the results are included in the appendix of the final paper.
> >
> > With all my major concerns addressed. I would be glad to increase my score to **strong accept**.

---

> > > ### Author Response · Authors · 2022-08-27
> > > **Thank you! Glad that you are willing to raise your recommendation!**
> > >
> > > Dear reviewer, thank you for your response! We are glad that you are willing to raise your recommendation!
> > >
> > > As you have suggested, we will attempt to include additional experiment results before the camera-ready deadline for CoRL if accepted.

---

> ### Author Response · Authors · 2022-08-18
> **Response to Review T7V5 (2/2)**
>
> > However, I am curious whether this current framework can encourage the task policy to fail at tasks in a “creative” way that can not be caught by the IRF policy. Since the IRF policy is only trained once with the VICE policy, it will not be able to adapt to other failure modes that the updated task policy induces. Although it seems like it is not the case in the experiments shown in the paper, I am wondering if the authors have observed such a phenomenon at all, and if the authors have any intuition on what to do in such cases. Minor point: Since the IRF policy is only trained once, it would be a little misleading to call it “GAN-like adversarial critic”.
>
> We did not observe any such “creative” or spurious behaviors from the task policy, even though it might seem reasonable to expect this at the outset, given what others have observed in adversarial learning settings. We have some hypotheses for why. In the standard "adversarial examples" settings, a neural network might easily and directly optimize its outputs to deliberately mislead a critic / discriminator network, such as by adding imperceptible noise to the pixels of an image. However, in our settings, creating such "adversarial perturbations" in the output is much harder because the actions of both the task policy and the interactive reward function (IRF) policy are mediated by the true physical dynamics of the environment. The space of sensible perturbations in the true environment state that might fool an IRF is much smaller, and achieving such perturbations is likely no easier than generating the correct task outcome.
>
> Furthermore, even if the updated task policy finds a shortcut to fool the IRF policy, updating the IRF policy iteratively (in a more GAN-like way) on the updated task policy’s data will close the exploit. As noted above, we have tried iterative training in early experiments and found it to be marginally beneficial.
>
> > Size of N: What is the size of N for the current experiments?
>
> Since we use a balanced sampling strategy, the size of N (negative actionable examples) is roughly equal to the size of P (positive actionable examples). We have updated Appendix A.6 to specify this. We attached the updated pdf in the response to the meta review.
>
> > Choice of hyperparameter: How sensitive is LIRF to the choice of λ?
>
> The IRF reward $\lambda D(o^*)$ applies only to the final state of the task policy, whereas the single-observation reward is computed densely for all states. (We have updated the Algorithm block to reflect this more clearly to avoid confusion on this point.)
>
> As such, $\lambda$ must be large for the IRF reward to sufficiently influence task policy learning, but beyond this, we do not believe that LIRF is very sensitive to the choice of $\lambda$: we picked 1000 heuristically. We will report results with other values of $\lambda$ within the response period.
>
> > Block stacking: From the quantitative results, as well as the number of positive examples needed, it seems like the block stacking experiment is most challenging. However, according to the appendix, the authors already made various attempts to simplify it, such as using motion primitives, and provide weight information once a block is picked up. Can the authors provide more explanation on why this experiment is still so challenging?
>
> In the weighted block stacking case, learning the optimal verification behavior of fine-grained poking is non-trivial. The IRF policy needs to simultaneously learn exactly what height to poke the block tower at, and with how much force. The tower is sensitive to perturbations - too forceful a poke at the wrong location could topple it even if it is in fact a “good” stack, with all the blocks in the right order. In summary, the IRF policy must learn to apply enough force to the tower to reveal its weight information, but be gentle enough to avoid toppling the tower. This “goldilocks” region of poking actions is hard to find, hence why it took 10,000 episodes to train a good IRF policy.
>
> > Typo?: “The valve can rotate freely without limits when the motor is not engaged, whereas it is locked in place or controllable through the motor when the motor is not engaged.” Should it be “the motor is engaged” near the end?
>
> Thank you for pointing this out. It is fixed in the updated paper.
>
> > Screwing task success: Can the authors provide more information on how the task success rate is evaluated in the screwing experiment? It is not very clear what does “the valve reaching the tightened state” exactly mean.
>
> We define the tightened state as when a designated spoke of the valve (the blue valve in Figure 3) is rotated 180 degrees past its starting orientation (aligned with the white line on the valve base). When this happens, the motor is engaged so the valve becomes immobile, simulating the “screwed” state.

---

> ### Author Response · Authors · 2022-08-23
> **Additional Experiment Results for Reviewer T7V5**
>
> We have completed the experiments proposed in our previous response, namely
> * Door closing - evaluating LIRF in a setting where VICE already achieves near-perfect performance
> * Ablation study on $\lambda$ .
>
> The results for the door closing experiment are shown below.
>
> | Policy |  Door Closing Success Rate |
> |---|---|
> | VICE | 100% |
> | LIRF | 99% |
>
> The VICE policy achieves near-perfect performance on the door closing task as it is fully-observed and relatively easy to learn. We can see that the performance of LIRF does not degrade under such circumstances.
>
> We also carried out ablation studies on the hyper-parameter $\lambda$, which is the weight for the IRF reward, in the door locking setting.
>
> | LIRF Policies |  Door Locking Success Rate |
> |---|---|
> | $\lambda$ = 10 | 32% |
> | $\lambda$ = 100 | 54% |
> | $\lambda$ = 1000 | 64% |
> | $\lambda$ = 10000 | 62% |
>
> As the IRF reward is only provided sparsely for the terminal state of the task policy, and the original single-observation reward $D(o_t)$ is provided for each step in the episode, we need $\lambda$ to be large enough in order to have a substantial effect during LIRF training. Beyond that, the performance of LIRF is not very sensitive to $\lambda$.

---

### Official Review · Reviewer_FXQi · 2022-07-31

**Originality:** Good
**Technical Quality:** Excellent
**Clarity Of Presentation:** Excellent
**Impact:** 4

**Recommendation:**

Strong Accept: I recommend accepting the paper and will argue for my recommendation even if other reviewers hold a different opinion.

**Summary:**

This paper introduces a new framework for interactive reward functions (IRF), which are a variant of exemplar rewards for scenarios where the relevant part of the state is not directly observable and needs to be verified through interaction. Each IRF is a separate policy trained with the goal of being able to discriminate positive demonstrations and negative examples from an untrained task policy. To train the task policy, after each task policy rollout, the IRF is executed to check the resulting state for task success. The framework is evaluated on three interesting showcases (door locking, block stacking, real-robot screwing) where clear advantages over vanilla exemplar rewards are shown. The framework is valuable for many real-life tasks where the task success is not camera observable.

**Issues:**

I do not have any other comments on any issues with this paper and congratulate the authors for a paper very well written.

**Quality Of The Limitations Section:**

Limitations are addressed clearly

**Reviewer Expertise:**

4: The reviewer is confident but not absolutely certain that the evaluation is correct

**Robotics Focus:**

Sufficient demonstration on hardware

**Strengths And Weaknesses:**

The justification and ideas of the paper are presented very clearly and in excellent writing. The experiments do a great job of showcasing the kind of task that such a framework is useful for.

The main contribution of this paper is this idea that the reward function be computed over an observation not from the end of the task policy rollout but instead one obtained by additionally using a separate policy. Indeed, for the simulation-based cases, it would be possible to directly train a success/failure discriminator on the (latent) state of the simulation at the end of the task policy, not requiring an additional reward function policy. It is only valuable for the (more important) real-world RL case where such information is not accessible. This is not immediately obvious since much of the experiments focus on simulation.

**Summary Of Recommendation:**

The paper does a great job of setting up this framework of interactive reward functions, describing how to train and use them, and showing experimentally their benefits over exemplar rewards. The setup is very useful for many real-world tasks since the partial observability situation applies very often.

---

> ### Author Response · Authors · 2022-08-18
> **Response to Review FXQi**
>
> > The paper does a great job of setting up this framework of interactive reward functions, describing how to train and use them, and showing experimentally their benefits over exemplar rewards. The setup is very useful for many real-world tasks since the partial observability situation applies very often
>
> Thank you for your efforts to evaluate and improve our manuscript, and for your kind positive feedback!
>
> > The only possible weakness is novelty, in that training a policy to check for task success is incremental enough an idea over exemplar rewards that even though a simple literature review does not produce other use cases, it is difficult to believe it has not been previously attempted, and unclear if this publication adds much value.
>
> We have thus far not been pointed to, nor otherwise encountered any prior work that has proposed the ideas in our submission, and we would argue that it would not be surprising for our work to be the first to attempt this. We request the reviewer to consider that good research projects often look obvious in retrospect (the well-documented “hindsight bias”); we believe that getting to LIRF from the current state of the art is non-trivial and involves a few significant steps. First, note that the majority of past work on robot learning treats the environment as fully observed, even if the observations are visual or otherwise partially observed [1,2,3,4]. Indeed, in practice, these methods have had success even in partially observed settings, through training with state-based reward functions (as our GT State Reward “baseline” approach does). In this context, even recognizing that exemplar rewards suffer from a specific weakness for partially observed settings is itself non-trivial. Subsequently, our specific solution involves developing a new abstraction of positive and negative "actionable" examples, and then devising an adversarial-style algorithm that trains an evaluation policy to improve a task policy trained from an aliased reward. We believe that none of these steps is obvious, and indeed, we ourselves tried other alternative approaches in the early stages of this project before we arrived at LIRF.
>
> [1] Bojarski, Mariusz, et al. "End to end learning for self-driving cars." arXiv preprint arXiv:1604.07316 (2016).
>
> [2] Akkaya, Ilge, et al. "Solving rubik's cube with a robot hand." arXiv preprint arXiv:1910.07113 (2019).
>
> [3] Kalashnikov, Dmitry, et al. "Scalable deep reinforcement learning for vision-based robotic manipulation." Conference on Robot Learning. PMLR, 2018.
>
> [4] Chebotar, Yevgen, et al. "Actionable models: Unsupervised offline reinforcement learning of robotic skills." arXiv preprint arXiv:2104.07749 (2021).

---

### Official Review · Reviewer_2X1d · 2022-08-02

**Originality:** Good
**Technical Quality:** Very Good
**Clarity Of Presentation:** Excellent
**Impact:** 2

**Recommendation:**

Strong Accept: I recommend accepting the paper and will argue for my recommendation even if other reviewers hold a different opinion.

**Summary:**

This paper introduces Learning from Interaction Reward Functions (LIRF), which addresses the problem of how to learn control policies under state aliasing, that is when success states look similar to failure states (e.g. a door locking problem). LIRF trains both a control policy to solve the task and an interactive reward function (another policy) to verify if the task has been completed correctly.

The main contributions of this work are
* The LIRF algorithm. Specifically, the idea to learn a verification policy from demonstrations.
* Experiments in simulation and on a physical robot which demonstrate how LIRF can be applied to solve 3 different robot learning tasks
* Experiments that show that IRFs can be reused during task execution to improve task performance


**Issues:**

* Please clarify if the other reward component, D(oT), is sparse
* Please give some intuition about why λ is so large
* It would be helpful if the authors could share their thoughts about how this approach might scale.
* Please include a discussion comparing LIRF with on inverse reinforcement learning in the related work
* Typo: line 226: experiment → experiments


**Quality Of The Limitations Section:**

Additional details required

**Reviewer Expertise:**

3: The reviewer is fairly confident that the evaluation is correct

**Robotics Focus:**

Sufficient demonstration on hardware

**Strengths And Weaknesses:**

Strengths
* The paper is very well written. It is lucid with a good selection of detail for the main paper. The introduction and motivation are especially clear.
* The paper addresses a relevant problem in robot learning — how to successfully train policies when success states look visually similar to failure states, i.e. when there is ambiguity due to state aliasing
* The idea of interactive reward functions is interesting and the idea of switching between a learned control policy and a learned verification policy is to my knowledge quite novel
* The experimental results are mostly convincing. I do have a couple of questions which I include in the following section. Additional comments.
   * LIRF is tested on two tasks in simulation and 1 task on a physical robot. The baselines appear to be fairly comprehensive and well chosen to shed light on different aspects of the problem
   * LIRF outperforms the main baseline VICE in all tasks. Since LIRF is VICE + interactive reward functions (IRF) I also appreciated how this comparison highlights the effect of IRF.
   * LIRF also outperforms Manual IRF in the simulated settings, indicating that learning IRFs has value (instead of manually defining them)
* I especially appreciate the efforts to demonstrate results on a physical robotic system and the use of a motor to “lock” the screw is nice
* The limitations section is substantive.

Weaknesses
* Experimental results
   * Was the GT State Reward permitted retries? I’m curious what the results would be if this policy has the same potential number of retries
   * The heuristically chosen λ value of 1000 seems quite high. I’m curious if the authors can give any intuition as to why it is so high? It would also be interesting to see an ablation over the value of λ. Further, given the weight is so high, I’m curious if the other reward component D(oT) can be removed altogether?
      * Relatedly, I’m curious if the original classifier reward is sparse (the IRF reward is sparse, lines 228-229)? If it was dense, perhaps this explains why it is important to include it even with a relatively low weight.
   * Lines 120-121 “if trained well with the right reward function, such history-based POMDP policies can still learn good task behaviors.” It would have been interesting to see how GT State Reward + a history-based policy performed on this task. Since all the tested policies are MDPs, including a GT State Reward + a history-based policy would also clearly show the effect of observation history compared to a perfect reward function in this case.
* LIRF seems very similar to inverse reinforcement learning (IRL) however IRL is not mentioned in the paper. Whilst LIRF learns a policy for a reward function for just a subset of the trajectory (the terminal state) and IRL learns it for all states, the two approaches appear to have essentially the same formulation. In both cases a reward function is learned from demonstrations which is in turn used to train an RL policy. I would have liked to see a discussion comparing the two in the related work.
* I wonder how well this approach would scale. For example, the 10k demonstrations required to learn the IRF for the block stacking tasks seems costly to replicate for other similarly complex tasks.
   * One advantage of exemplar rewards is that just a single image is required instead of a demonstration trajectory. LIRF doesn’t have this benefit since it requires verification trajectories. In some cases these may require a similar amount of effort to full demonstrations.
* I do appreciate the substantial limitations section. However, related to the above it does seem that the time required to collect (and not just store) the positive actionable examples could be a limitation (e.g. if 10k per task is often required). I would have liked to see a discussion about this in the work.


**Summary Of Recommendation:**

This paper presents an interesting and novel approach to a relevant problem in robot learning, solving tasks when there is state aliasing. It is well written throughout and the experiments in simulation and the real world demonstrate the effectiveness of the proposed method LIRF. I do think a comparison with IRL is missing and I have concerns about how scalable the method is given the requirement to gather potentially thousands of verification trajectories. However, on balance the strengths outweigh the weaknesses.

---

> ### Author Response · Authors · 2022-08-18
> **Response to Review 2X1d (1/3)**
>
> We thank reviewer 2X1d the effort spent to evaluate our manuscript and provide feedback!
>
> **Clarification on LIRF and “demonstrations”:** Before addressing their specific questions, we would like to note first that we have noticed a possible misunderstanding of our key contributions, that informs several of these questions. To clarify, our approach ***LIRF does not in fact require demonstrations*** either of how to perform a task, or of how to verify that a task is complete. It instead acquires both these behaviors from a task specification that consists purely of “actionable positive examples”, which are examples of successful task outcomes. For example, for a table construction task, this means providing examples of well-built tables, rather than recording full demonstrations of a teleoperated robot assembling tables (as in imitation or inverse reinforcement learning), or even recording full demonstrations of a teleoperated robot inspecting an assembled table, such as by tugging at its legs etc.
>
> >LIRF seems very similar to inverse reinforcement learning (IRL) however IRL is not mentioned in the paper. Whilst LIRF learns a policy for a reward function for just a subset of the trajectory (the terminal state) and IRL learns it for all states, the two approaches appear to have essentially the same formulation. In both cases a reward function is learned from demonstrations which is in turn used to train an RL policy. I would have liked to see a discussion comparing the two in the related work.
>
> While loosely related, LIRF is in fact substantially distinct from inverse reinforcement learning (IRL).  We briefly mentioned IRL in the paper (L80), and we will expand here to clarify the distinction in two stages, separating IRL first from exemplar rewards, and then from our *interactive* reward functions-based LIRF approach.
>
> First, IRL requires full state-action demonstrations from an expert policy to learn a reward function r(s,a,s’), such as by comparing against the distribution of (s, a, s’) transitions observed in demonstrations. Exemplar rewards are much less demanding, requiring only examples of successful task *outcomes* rather than demonstrations. Further, learning purely from goal specifications means that the agent goes beyond merely imitating provided solutions to *discover* task solutions, which is a core desideratum of intelligent behavior. Finally, it is possible to make the case that IRL is a special, task-supervision-intensive case of exemplar rewards, where the policy must achieve not just task outcomes but all intermediate states as seen in expert demonstrations (see Sec 5 in Fu et al [1]).
>
> Our approach LIRF goes further than merely using exemplar rewards however. Indeed, the key idea of LIRF is to extend exemplar reward functions to *interactive exemplar reward function* policies that **interact with the environment** to infer the reward for a task outcome. In this way, LIRF differs from both IRL and exemplar rewards which only learn standard “passive” reward functions r(s,a,s’).
>
> [1] Fu, Justin, et al. "Variational inverse control with events: A general framework for data-driven reward definition." Advances in neural information processing systems 31 (2018).

---

> ### Author Response · Authors · 2022-08-18
> **Response to Review 2X1d (2/3)**
>
> > “The LIRF algorithm … learns a verification policy from demonstrations.”
>
> > “I wonder how well this approach would scale. For example, the 10k demonstrations required to learn the IRF for the block stacking tasks seems costly to replicate for other similarly complex tasks. One advantage of exemplar rewards is that just a single image is required instead of a demonstration trajectory. LIRF doesn’t have this benefit since it requires verification trajectories. In some cases these may require a similar amount of effort to full demonstrations. I do appreciate the substantial limitations section. However, related to the above it does seem that the time required to collect (and not just store) the positive actionable examples could be a limitation (e.g. if 10k per task is often required). I would have liked to see a discussion about this in the work.”
>
> We would like to reiterate: unlike, say, IRL, **LIRF requires no demonstrations either of the task policy or of the verification behaviors**. Instead, LIRF learns both these policies from a task specification that consists purely of actionable positive examples. As mentioned above, examples are less cumbersome to gather than demonstrations since the expert only needs to show the final outcome of the task, not how to solve the task. Further, even in an application setting that requires these positive examples to be constructed on the fly to present to LIRF during training, they could be constructed by a different agent (such as a human supervisor) rather than by using the embodiment of the robot learner, as most practical imitation learning approaches require.
>
> All this said, we agree that there are cases in which the number of *actionable positive examples* required by LIRF is large, and gathering and storing them is non-trivial, our weighted block stacking task being an example. We point this out concisely in the limitations section for space as you have noted, and discuss this further below.
>
> **First, what causes the number of required actionable positive examples to be high or low?** In the weighted block stacking task, the desired task outcome, a well-constructed stack of blocks, is not objectively a very stable object configuration (even if it is more stable than poorly constructed stacks) — in particular, relatively small perturbations can destroy it. As a result, we use ~7k positive examples, over 10k IRF training episodes (the remaining ~3k episodes don’t result in the destruction of a positive example so that we can reuse the examples provided).  Further, it takes a long time for the interactive reward function (IRF) to learn the nuanced poking behavior that can accurately separate “good” block stacks from “bad”, to then produce a good task policy. The IRF policy must apply enough force to the tower at the right locations to reveal the weights of the blocks, but be gentle enough to avoid toppling the tower. On the other extreme, in the screwing task, we required only one single positive example of a tightened screw, over 100 IRF training episodes with positive samples: here, the perturbation behavior to separate loose and tight screws is very straight-forward to learn, and the positive examples are not easily destroyed. (more on this below)
>
> **Second, how should we think about the cost of training LIRF policies in these terms?** When the IRF policy does not destroy a positive example, it can be reused. For example, in the screwing task, technically only **one single positive example** of a tightened screw is required because the positive example physically cannot be destroyed (undoing a tight screw) by the robot during training, so that this one tightened screw can be reused for 100 times. Note that in the paper, we reported 250, 10k and 100 trials with positive samples for training IRF policies for door locking, weighted block stacking, and screwing tasks respectively. However, if we take reusability of positive examples into consideration, the IRF training only requires 110, 7k, and 1 positive examples.
>
> **Finally, how could LIRF be encouraged to learn from fewer positive examples?** Here, we speculate that explicitly penalizing the destruction of positive examples and/or explicitly encouraging the verification policy to generate reversible behaviors will help. Further, we have used off-the-shelf policy learning approaches (SAC) without optimizing for sample efficiency, and there may be significant gains from using alternative, more sample-efficient approaches, such as from the model-based reinforcement learning literature.
>
> We are happy to add this discussion to the appendix if suggested.

---

> ### Author Response · Authors · 2022-08-18
> **Response to Review 2X1d (3/3)**
>
> > Was the GT State Reward permitted retries? I’m curious what the results would be if this policy has the same potential number of retries
>
> The ground truth (GT) state reward baseline doesn't directly permit retries — recall that rewards in RL are typically assumed available only at training time, and the task policy for this baseline is assumed to only have access to observation histories (same as all the other baselines and our approach).  We propose to instead evaluate two alternative approaches in the spirit of this suggestion:
> * First, we will run the GT state reward baseline with increased episode length so that its total duration matches the K retries the LIRF policy gets.
> * Second, we will provide ground truth state reward access to check for task success instead of our IRF reward policy. This GT state reward policy with GT state reward checking would correspond to an “oracle” version of our method.
>
> We have begun work towards running these experiments and will aim to post the results within the response period. Please let us know if you would suggest any changes. And finally, note that neither of these are really “baselines” that are fair to compare LIRF against — each enjoys significant advantages in accessing the true state to varying extents for task specification and verification.
>
> > The heuristically chosen λ value of 1000 seems quite high. I’m curious if the authors can give any intuition as to why it is so high?
> Further, given the weight is so high, I’m curious if the other reward component D(oT) can be removed altogether?
>
> Note that when training LIRF policies, just as when training VICE policies, we apply the original single-observation reward D(o_t) for *each step in the episode* (100 steps for door locking and screwing, 20 steps for block stacking), and add the IRF reward as a sparse bonus reward only for the terminal state of the task policy. This is why the lambda value is set heuristically to be so high, to permit the IRF reward to have a substantial effect even though it is provided much more sparsely. We are now experimenting with other values of lambda, and will aim to report the performance within the response period.
>
> We recognize that this question may have arisen because the above explanation isn’t explicit in the main paper. Thank you for bringing this to our attention, we have now updated Algorithm 1 with the full reward description:
> where we use the IRF reward in the terminal state $\hat{R}(o_T) = D(o_T) + \lambda D(o^*) $ and  $\hat{R}(o_t) = D(o_t)$ for all intermediate states $o_t$.
>
> > Relatedly, I’m curious if the original classifier reward is sparse (the IRF reward is sparse, lines 228-229)? If it was dense, perhaps this explains why it is important to include it even with a relatively low weight.
>
> In our early experiments, we found that it helped to retain the dense single-observation rewards when training LIRF task policies; it substantially eases exploration during learning, as might be expected.
>
> > Lines 120-121 “if trained well with the right reward function, such history-based POMDP policies can still learn good task behaviors.” It would have been interesting to see how GT State Reward + a history-based policy performed on this task. Since all the tested policies are MDPs, including a GT State Reward + a history-based policy would also clearly show the effect of observation history compared to a perfect reward function in this case.
>
> Indeed, our GT State Reward baseline uses history-based policies for the screwing and weighted block stacking task (For each task, all compared methods in Table 1 use the same inputs to their respective policies). We had pointed this out in Appendix A.4.3, but have now updated the experiment section in the main paper to make this clear, thank you again. To restate the conclusions, GT State Reward performs slightly worse than our LIRF+Verify policies on average, and performs better than plain LIRF policies, with the advantage of having access to ground truth state-based task rewards.

---

> ### Author Response · Authors · 2022-08-23
> **Additional Experiment Results for Reviewer 2X1d**
>
> We have completed the experiments proposed in our previous response for the door locking setting, namely
> * GT state reward baseline with increased episode length so that its total duration matches the K retries the LIRF policy gets.
> * GT state reward policy with GT state success checking.
> * Ablation study on $\lambda$ .
>
> The results for GT state reward baselines with multiple retries are shown below.
> | Policy |  Door Locking Success Rate |
> |---|---|
> | GT state reward with 3 retries | 71% |
> | GT state reward with 10 retries | 60% |
> | GT state reward with 3 retries and GT state success checking | 91% |
> | GT state reward with 10 retries and GT state success checking | 98% |
>
> The experimental results align with our expectation that, without GT state success checking, the performance of GT state reward baselines does not improve with more retries since they tend to overturn the door latch. With GT state success checking, the GT state reward policies with multiple retries stop right after they successfully lock the door, achieving better performance, as they serve as an “oracle” version of our method.
>
> We also carried out ablation studies on the hyper-parameter $\lambda$, which is the weight for the IRF reward, in the door locking setting.
>
> | LIRF Policies |  Door Locking Success Rate |
> |---|---|
> | $\lambda$ = 10 | 32% |
> | $\lambda$ = 100 | 54% |
> | $\lambda$ = 1000 | 64% |
> | $\lambda$ = 10000 | 62% |
>
> As the IRF reward is only provided sparsely for the terminal state of the task policy, and the original single-observation reward $D(o_t)$ is provided for each step in the episode, we need $\lambda$ to be large enough in order to have a substantial effect during LIRF training. Beyond that, the performance of LIRF is not very sensitive to $\lambda$.

---

> > ### Comment · Reviewer_2X1d · 2022-08-26
> > **Thank you for your detailed response.**
> >
> > Thank you for your detailed response.
> >
> > I appreciate the extensive clarification of LIRF compared with IRL and, given the differences, I agree with the author's decision not to discuss it more in the paper. I acknowledge I had misunderstood what an actionable positive example was and do see that LIRF requires no demonstrations either of the task policy or of the verification behaviors.
> >
> > Relatedly I also appreciate the discussion on the scalability of LIRF and it would be great if the authors’ could add this to the appendix since I think it would be valuable for other readers. The authors’ ideas to encourage policies not to destroy the positive and / or to learn to reverse destructive behaviors is really interesting. I hope to see this in future work but agree it is out of scope for this paper.
> >
> > Finally, I appreciate the authors efforts to clarify the GT State reward policy architecture and for running additional experiments. The ablation study on the size of λ is interesting and given the dense single observation reward it makes sense that λ needs to be large. The additional results for the GT State reward baseline are also interesting. It suggests that allowing policies to retry when they fail could be beneficial (since LIRF+Verify > LIRF and GT State Reward + State success checking > GT State reward).
> >
> > In summary, the authors have fully responded to my concerns and questions and I will change my recommendation to a Strong Accept.

---

> > > ### Author Response · Authors · 2022-08-26
> > > **Thank you for your timely response! Glad that you are willing to raise your recommendation!**
> > >
> > > Dear reviewer 2X1d, thank you for your timely response! We are glad that you are willing to raise your recommendation to "strong accept"!

---

### Official Review · Reviewer_BeWt · 2022-08-09

**Originality:** Very Good
**Technical Quality:** Very Good
**Clarity Of Presentation:** Very Good
**Impact:** 4

**Recommendation:**

Strong Accept: I recommend accepting the paper and will argue for my recommendation even if other reviewers hold a different opinion.

**Summary:**

This paper presents a method for extending exemplar rewards, where reward is inferred from success examples, to the partially observable setting. Existing methods don't extend well since they match the current observation to a success observation, but the observation may not be sufficient for determining success.

The paper proposes LIRFs. A task policy is pretrained with existing exemplar reward methods, then rolled out to get negative (but reachable) outcomes. Then, an IRF policy is trained to start in either an exemplar or a negative outcome and perturb the environment until it can label success/failure. This RL approach, rather than a classification approach, is said to avoid use of full trajectories.

Evaluation includes three tasks/action spaces and SAC for both policies. LIRF beats full-obs IRF methods deployed in PO case and loses only to methods with privileged info. Using LIRF during task execution (not just as reward) to check success beats all baselines.



**Issues:**

- As I said in "weaknesses", the presentation in section 5 should move from 1) lots of space devoted to intuition and 2) laundry lists of facts to 1) most of that space devoted to analysis (even ablations) and comparison, and 2) claims-driven writing

**Quality Of The Limitations Section:**

Limitations are addressed clearly

**Reviewer Expertise:**

4: The reviewer is confident but not absolutely certain that the evaluation is correct

**Robotics Focus:**

Sufficient demonstration on hardware

**Strengths And Weaknesses:**

**Strengths**:
- Paper is clear and well written
- The proposed approach is creative. The choice to use RL for pi_R is nonobvious, so it's satisfying that it provides clear value in terms of efficiency.
- Evaluation has been done fairly thoroughly, and the baseline choices are thoughtful and probably complete for the problem at hand even if not very extensive (I'm not personally familiar with other state of the art exemplar reward methods anyway).

**Weaknesses**:
- It may be worth comparing briefly to a broader range of methods, to give a good argument for exemplar rewards as a whole (the metric doesn't have to be accuracy, it can be data cost for example). However, I would defer to other reviewers that have read more papers in the particular space than I have.
- The paper is a great read, but section 5 becomes a bit dry. The paragraph titles are promising - it's very pedagogical to offer readers intuition for each module and I appreciate that that theme holds throughout the paper. However, we don't need such a play-by-play as we're given in 5.1 - the second and third paragraph can be one brief paragraph.
- Following on that, it would be helpful to fill the saved space with more analysis of the results. Right now the paper seems to argue that LIRFs exist in a category of their own, and I buy that, but the result analysis is a bit dry and laundry list-like, and could be more claims-driven.


**Summary Of Recommendation:**

I think this paper hits all the bases it needs to. I would like more insight in analysis, but it's a solid, successful, and (to my knowledge) novel work. I don't see a reason not to accept it.

---

> ### Author Response · Authors · 2022-08-18
> **Response to Review BeWt (1/2)**
>
> We thank reviewer BeWt for the thorough feedback! We are glad that the reviewer recognized our method’s novelty.
>
> > It may be worth comparing briefly to a broader range of methods, to give a good argument for exemplar rewards as a whole (the metric doesn't have to be accuracy, it can be data cost for example). However, I would defer to other reviewers that have read more papers in the particular space than I have
>
> Thank you. The general case for exemplar rewards versus alternative forms of task specification (such as engineered reward functions, inverse reinforcement learning, and learning from human feedback) is stated briefly in Sec 1 L17-31 & Sec 2.1 L74-83 in our paper, and before that, in detail in prior works [1-4]. We briefly restate these arguments below. Next, we reframe the experimental results from the paper more clearly in terms of comparing example-based rewards and LIRF-style actionable example-based rewards versus various alternative forms of task specification.
>
> **The conceptual case for exemplar rewards:** The current standard approach for task specification in RL is to engineer reward functions, but this can be cumbersome and prone to misspecification. It is common for reward functions to contain up to dozens of heuristic terms with weights tuned from trial and error, and to rely on privileged task-specific sensors to gather true task state information. To circumvent such laborious manual reward design, several families of approaches aim to learn rewards from data. For example, imitation learning and inverse reinforcement learning methods learn policies and task rewards from optimal demonstrations, but such demonstrations are typically expensive or may be impossible to obtain, and further often require human expertise for manually operating a robot to demonstrate how to solve a task. Other methods train RL agents by learning from human feedback but have the drawback of requiring in-the-loop query-able human teachers [6, 7].
>
> Against this backdrop, exemplar rewards, proposed relatively recently [1-3], is a particularly promising new task specification framework that avoids collecting demonstrations or engineering reward functions, by instead requiring only example success outcomes for task specification. This does away with the need for human expertise either for reward engineering or for demonstration specification, and permits a versatile, simple, and purely offline task specification that states *what* the goal is without also spelling out *how* to achieve it.
>
> **Empirical comparisons:** Besides this conceptual distinction, we also take care in our evaluations (see Tab 1) to study a range of comparable alternatives to actionable examples for task specification: goal observation examples (VICE), state demonstrations (GAIfO), human engineered interaction (Manual IRF), and ground truth state-based engineered rewards (GT State Reward). The relative costs of these different task specifications are task-dependent, and hard to compare fully objectively, but GAIfO, Manual IRF, and GT State Reward, all arguably require more cumbersome environment instrumentation, reward engineering, and/or data collection than our approach LIRF in most realistic task settings, and we find that our approach outperforms all of them in terms of task performance on average.
>
> It is hard to be fully comprehensive in evaluating all alternative forms of task specification, but we could certainly add more here, if the reviewer believes it would be informative. As one example, we could compare against, say, generative adversarial imitation learning (GAIL) [5] from state-action demonstrations (as opposed to state-only demonstrations in GAIfO). This would involve considerably more supervision / “data cost” than LIRF, and arguably more than all the methods in Table 1. However, if the reviewer deems this useful, we will aim to produce preliminary results for this within the response period. Please let us know.

---

> ### Author Response · Authors · 2022-08-18
> **Response to Review BeWt (2/2)**
>
> >The paper is a great read, but section 5 becomes a bit dry. The paragraph titles are promising - it's very pedagogical to offer readers intuition for each module and I appreciate that that theme holds throughout the paper. However, we don't need such a play-by-play as we're given in 5.1 - the second and third paragraph can be one brief paragraph.
> Following on that, it would be helpful to fill the saved space with more analysis of the results. Right now the paper seems to argue that LIRFs exist in a category of their own, and I buy that, but the result analysis is a bit dry and laundry list-like, and could be more claims-driven.
>
> Thank you for this suggestion to revise how our experiments are presented, we agree. Briefly, our experiments answer the following three questions in the affirmative: (1) Does our interactive reward learning framework enable us to learn partially observable tasks that other baselines cannot? (2) Does the learned IRF policy sensibly evaluate task completion? (3) Can using the IRF policy in-the-loop for task verification improve task execution? We have revised the description of experiments and results in Sec 5 to more clearly state these questions and how they are answered. We attached the updated pdf in the response for the meta review.
>
> [1] Fu, Justin, et al. "Variational inverse control with events: A general framework for data-driven reward definition." Advances in neural information processing systems 31 (2018).
>
> [2] Eysenbach, Benjamin, et al. “Replacing rewards with examples: Example based policy search via recursive classification.” Advances in Neural Information Processing Systems, 34 (2021).
>
> [3] Singh, Avi, et al. "End-to-end robotic reinforcement learning without reward engineering." arXiv preprint arXiv:1904.07854 (2019).
>
> [4] Xie, Annie, et al. "Few-shot goal inference for visuomotor learning and planning." Conference on Robot Learning. PMLR, 2018.
>
> [5] Ho, Jonathan, and Stefano Ermon. "Generative adversarial imitation learning." Advances in neural information processing systems 29 (2016).
>
> [6] Christiano, Paul F., et al. "Deep reinforcement learning from human preferences." Advances in neural information processing systems 30 (2017).
>
> [7] Akinola, Iretiayo, et al. "Accelerated robot learning via human brain signals." 2020 IEEE international conference on robotics and automation (ICRA). IEEE, 2020.

---

> > ### Comment · Reviewer_BeWt · 2022-08-27
> > **Changes look good**
> >
> > Hi,
> >
> > I think the changes to section 5 are helpful and clearer. In response to the below comment, I think those comparisons would be helpful but the paper is already ready for acceptance. I think the efforts in response to the other reviewers have boosted the apparent significance, which was my primary concern. I'll maintain my existing Strong Accept rating.

---

> > > ### Author Response · Authors · 2022-08-27
> > > **Thank you!**
> > >
> > > Dear reviewer, thank you for this response.
> > >
> > > As you have suggested, we will attempt to include additional comparisons to alternative forms of task specification before the camera-ready deadline for CORL if accepted.

---

### Meta-Review · Area_Chair_KMnG · 2022-08-15

**Recommendation:** Accept (Oral)
**Confidence:** 4

**Metareview:**

The reviewers have unanimous decision to accept this paper. I thank the authors for their very thorough comments and answers.

Summary: The paper seems to be well motivated and clearly written. All reviewers left a very detailed review.

Quality(5/5): All reviewers agree that the paper is very well-written and technically convincing.

Clarity(5/5): All reviewers agree that the paper is very clear. But one reviewer suggested that Section 5 has to be more polished.

Originality(5/5): No one questioned the originality of this work.

Significance (3/5): I see different opinions among the reviewers. The reviewers generally agree that interactive reward is new and creative. However, they have mixed opinions on the impact of this work. The reviewer "2X1d", "FXQi" and "T7V5" mentioned multiple important weaknesses of this paper. They should be addressed or clarified well before acceptance. The question about 1) comparisons to IRL  by 2X1d, 2) assumption on VICE performance by T7V5 seem particularly important.

**Best Paper Nomination:**

Yes

---

> ### Author Response · Authors · 2022-08-18
> **Response to Meta Review KMnG**
>
> Dear Area Chair, thank you for your efforts in evaluating our submission.
>
> We believe we have addressed the main concerns of each reviewer, or are engaging with them by discussing and running followup experiments. Here, we highlight the two key points as identified in your comment on significance above.
>
> **Comparisons to IRL:** Reviewer 2x1d expressed concern about the relationship between our method LIRF and inverse reinforcement learning (IRL), but we believe this may have been caused primarily by misunderstanding our method as relying on demonstrations. In reality, LIRF trains both task and verification policies based only on success examples as task outcomes. We have clarified this extensively. First, as stated above, LIRF only requires task outcome examples, unlike IRL methods that rely on full state-action trajectory demonstrations of how to accomplish a task. Perhaps more importantly, LIRF’s key idea is to learn an *interactive reward function policy* that interacts with the environment to infer the reward corresponding to a task outcome; this differentiates it even more from IRL, which only learns passive reward functions $r(s, a, s’)$.
>
> **Assumptions on VICE Performance:** Next, we have clarified for Reviewer T7V5 that LIRF task policy performance does not in fact degrade if the initial VICE policy is high-performing, and relatedly, that LIRF can handle multiple iterations of training. We have reported an empirical experiment that validates our responses. Instead, all that can be said about LIRF’s gains over VICE is that they are expectedly larger in partially observed settings where VICE fails, and which LIRF is explicitly designed to handle.
>
> We have also responded to all other comments raised by reviewers, and hope that our engagement in the rest of the response period will help confirm that all concerns are addressed.
>
> Finally, we have updated the pdf of the paper and Appendix. Note that it is temporarily over the page limit to address all reviewers’ comments. We will make sure to edit the paper to fit into the page limit if accepted, moving text to appendices as appropriate based on reviewer responses.

---

> ### Author Response · Authors · 2022-08-27
> **Message to AC KMnG after Updating Additional Experiments**
>
> Dear area chair, we would like to thank you again for evaluating our submission.
>
> On Aug 23, we completed the experiments proposed in our first-round of responses for each reviewer, including ground truth ("GT") state reward baselines with multiple retries, an ablation study on hyperparameter $\lambda$, and evaluating our approach LIRF in a setting where VICE already achieves near-perfect performance. The results were posted on Aug 23 in our responses to Reviewer 2X1d and T7V5.
>
> Since posting our initial responses on Aug 19 and the experiments on Aug 23, we are glad that reviewer 2X1d has engaged with our responses and is willing to raise their recommendation to "strong accept". Unfortunately, we are yet to hear back from the remaining reviewers and are anxious to do so as the discussion deadline approaches. We hope that other reviewers will acknowledge that their concerns have been satisfactorily addressed, or otherwise raise any follow-up questions that arise while there is still time for us to respond.